# Pep27 Mutant Immunization Inhibits Caspase-14 Expression to Alleviate Inflammatory Bowel Disease via Treg Upregulation

**DOI:** 10.3390/microorganisms10091871

**Published:** 2022-09-19

**Authors:** Hamid Iqbal, Gyu-Lee Kim, Ji-Hoon Kim, Prachetash Ghosh, Masaud Shah, Wonsik Lee, Dong-Kwon Rhee

**Affiliations:** 1School of Pharmacy, Sungkyunkwan University, Suwon 16419, Korea; 2Department of Physiology, Ajou University, Suwon 16499, Korea; 3Research Center, DNBIO, Suwon 16419, Korea

**Keywords:** ∆pep27, inflammatory bowel disease, regulatory T cells, caspase-14, immune tolerance, gut microbiome

## Abstract

Inflammatory bowel disease (IBD) is a highly prevalent gut inflammatory disorder. Complicated clinical outcomes prolong the use of conventional therapy and often lead to compromised immunity followed by adverse events and high relapse rates. Thus, a profound medical intervention is required. Previously, intranasal immunization of pneumococcal *pep27* mutant (Δpep27) exhibited long-lasting protection against immune-related disorders. System biology analysis has predicted an inverse correlation between Δpep27 immunization and gastroenteritis. Recently, we established that Δpep27-elicited Tregs repressed Wnt5a expression and enhanced barrier integrity, suggesting the restoration of immunological tolerance. Therefore, we evaluated whether Δpep27 can alleviate IBD. Δpep27 dose-dependent response was analyzed in dextran sulfate sodium-induced mice using transcriptome analysis. Pro- and anti-inflammatory signatures were cross-correlated by quantitative PCR and western blot analyses. To address the hierarchy regulating the activity of caspase-14, an undefined marker in IBD, and regulatory T cells (Tregs), antibody-based neutralization studies were conducted. Fecal microbiome profiles were analyzed by 16S rRNA pyrosequencing. Δpep27 significantly attenuated dextran sulfate sodium-induced oxidative stress parameters, proinflammatory cytokines, caspase-14 expression level, and upregulated tight junction, anti-inflammatory genes IL-10 and TGF-β1 via upregulation of Tregs to restore healthy gut microbiota. Neutralization studies unveiled that ∆pep27 had a remedial effect via Treg upregulation. Caspase-14, being an important mediator in the pathogenesis of IBD, can be an alternate therapeutic target in IBD. ∆pep27-increased Tregs repressed caspase-14 expression and reversed gut microbial dysbiosis, aiding to re-establish immunological tolerance.

## 1. Introduction

Inflammatory bowel disease (IBD), such as Crohn’s disease and ulcerative colitis (UC), is characterized by uncontrolled chronic inflammation and epithelial barrier disruption caused by aberrant cytokine production and dysregulated immune response with a loss of immune tolerance. The etiology of IBD remains unclear; however, it is a multifactorial disease that involves a genetic predisposition and altered microbiota with inter-subject variability and relapses [1,2,3]. 

To overcome IBD, conventional methods have been utilized with anti-inflammatory regimen to achieve disease remission either by immune modalities such as corticosteroids, antibiotics, aminosalicylates, neutrophil-derived factors, and probiotics, or by neutralization of proinflammatory cytokines using antibodies (e.g., anti-TNF-α or IL-1β). However, the current regimen has several clinical limitations such as immunosuppression, low responsiveness, and even refractoriness [3,4]. The depletion of anti-inflammatory Tregs and the enrichment of proinflammatory Th17 cells have been linked to IBD pathogenesis [5,6,7]. As Tregs suppress Th17 inflammation and secrete IL-10 and transforming growth factor-β (TGF-β1) to induce immunotolerance [8,9,10], the administration of Treg-inducing strains and fecal microbiota transplantation have been developed to rehabilitate IBD [10,11,12]. As gut microbiota can be enriched and/or selected by diet, microbial implementation can be reversed by diet and result in IBD recurrence. Thus, an effective medical intervention is required. 

Growing evidence suggests that mucosal healing is an emerging therapeutic strategy that could result in clinical management of IBD [13]. The *pep*27 gene encodes an autolysis-inducing factor in *Streptococcus pneumoniae* (pneumococcus). During blood invasion, pneumococcal *pep27* is induced, whereas the *pep27* mutant (∆pep27) is devoid of blood invasion and becomes avirulent. Previously, we demonstrated that intranasal immunization with an attenuated non-invasive Δpep27 strain substantially protected against pneumococcal infections and influenza virus challenges and showed rapid clearance by 24 h post-intranasal immunization undetected in lung and blood, suggesting that Δpep27 could not invade into other tissues [14,15,16,17,18]. Moreover, Δpep27 demonstrated a preventive effect against asthma via Treg induction [19], and can significantly increase the survival rate against non-specific pathogens demonstrating long-lasting immunity [17]. Interestingly, system biology analysis predicted a protective role of Δpep27 in the lungs to inhibit the abnormalities of the large intestine (Figure 1A) suggesting the gut–lung axis is a bidirectional communication network. Recently, we investigated the nuclear factor of activated T cell (NFAT), a downstream molecule of the noncanonical Wnt signaling expression including Wnt5a and Wnt11 in a dextran sulfate sodium (DSS)-induced colitis (DIC) model, suggesting a potential proinflammatory role in the pathogenesis of IBD [20]. Moreover, Δpep27 immunization-induced Treg represses Wnt5a expression, and helps restoration of gut tolerance [20]. In this study, to confirm ∆pep27 as highly pragmatic vaccine, we further analyzed the potential role of caspase-14, an unidentified marker in IBD, Treg efficacy, and microbial pattern to elucidate the underlying mechanism. 

## 2. Materials and Methods

### 2.1. Experimental Animals

C57BL/6 female mice, aged 5–7 weeks, were purchased from Orient Bio, Korea and housed with free access to food and water at 22 °C with a 12 h/12 h light/dark cycle for 7 days to acclimatize prior to the commencement of the experiment. 

### 2.2. Induction of Colitis and Immunization Protocol

Mice were weighed, randomly divided, and subjected to the experimental IBD model in four groups: (1) Control, (2) DSS, (3) ∆pep27 + DSS, and (4) ∆pep27. Mice were intranasally immunized with ∆pep27, 3 times once a week with 1 × 10^7^ to 1 × 10^8^ CFU using 50 µL intraperitoneal ketamine (Figure 1B). Experimental colitis was induced by the addition of 4% (wt/vol) DSS (42867-100G, Sigma-Aldrich) in the drinking water for 7–9 days until sacrifice for all experimental groups.

### 2.3. Macroscopic Assessment of Colitis Severity

Progression of colitis was rated daily as described previously [21], and assigned a score based on criteria used to calculate an average disease activity index (DAI) on a daily basis as follows. Weight loss as 0, no change; (1) <5%; (2) 6–10%; (3) 11–20%; (4) >20%. General appearance as 0, normal; (1) piloerect; (2) lethargic, piloerect; (3) lethargic and hunched; (4) motionless, sickly. Stool consistency as 0, normal; (1) pasty, semiformed; (2) sticky; (3) sticky with blood; (4) unable to defecate. Rectal bleeding as 0, no blood; (1) negative hemoculture; (2) positive hemoculture; (3) visible blood on fur; (4) gross rectal bleeding.

### 2.4. Histopathological Analysis 

Colon sections were fixed overnight in neutral buffered formalin (10%) and stained with hematoxylin-eosin (H&E), according to the manufacturer’s instructions (GS Tech Korea Co., Ltd., South Korea). Images were obtained at 20× magnification with a light microscope (Olympus IX73, Olympus U-TB190, Tokyo, Japan) for histopathological examination using a previously established scoring system [21]. The remaining colon was frozen immediately in liquid nitrogen and stored at −80 °C for further analysis.

### 2.5. Terminal Deoxynucleotidyl Transferase-Mediated dUTP Nick-End Labeling Assay

Apoptosis in the colon tissue was quantified by assessment with a terminal deoxynucleotidyl transferase dUTP nick-end labeling (TUNEL) assay using a test kit according to the manufacturer’s instruction (GS Tech Korea Co., Ltd.). The apoptotic index was defined as the percentage of TUNEL-positive cells, as described previously [22]. Apoptotic cells were counted in a minimum of seven visual fields in each section under an inverted light microscope (200×) (Olympus IX73, Olympus U-TB190, Tokyo, Japan).

### 2.6. Assessment of Myeloperoxidase (MPO) Activity and Malondialdehyde (MDA) Level

To investigate the production of reactive oxygen species (ROS) and levels of reactive nitrogen species (RNS) as an indicator of neutrophil infiltration into the colon, myeloperoxidase (MPO) activity was quantified as described previously [23]. The level of malondialdehyde (MDA), an index of the peroxidation reaction, was evaluated as described previously [23]. 

### 2.7. Permeability Assay

The quantification of vascular permeability was based on the extent of an Evans blue (E2129-10G, Lot # MKCB2532V, Sigma-Aldrich) extravasation assay. Briefly, Evans blue dye was injected into anesthetized mice for 30 min, and then the colon was excised, weighed, homogenized in 1 mL PBS, and extracted overnight in 500 µL of formamide at 55 °C. The Evans blue concentration in the colon homogenate supernatants was quantified by spectrophotometric absorbance at 610 nm [24].

### 2.8. Gut Microbiome Analysis in Fecal Samples

DNA was extracted from fecal pellets by first agitating with garnet beads (MO BIO Laboratories, Carlsbad, CA, USA) for 10 min and then processed with a bacterial genomic DNA extraction kit (Thermo Scientific Korea, Seoul, Korea). Extracted DNA was amplified with primers targeting the V1 to V3 hypervariable regions of the bacterial 16S rRNA gene, and the amplicons were sequenced using a 454 GS FLX titanium pyrosequencer (Roche, Branford, CT, USA) and analyzed at Chunlab (Chunlab Inc., Seoul, Korea) as described previously [25]. The taxonomic classification of each read was assigned based on the EzTaxon-e database (http://eztaxon-e.ezbiocloud.net, accessed on 5 February 2021). The richness and evenness of samples were determined by several alpha diversity indicators including abundance-based coverage estimators (ACE), Chao1, and phylogenetic and Jackknife estimation. In addition, the Simpson diversity index and Shannon diversity index at 3% distance were calculated in the CL community program (Chunlab Inc.).

### 2.9. Treg and Caspase-14 Neutralization

To verify the effect of ∆pep27 immunization effect on Treg depletion [26,27], mice were intraperitoneally (i.p.) administered 100 µg of anti-CD25 antibody (Cat # 102058, Ultra-LEAF^®^ anti-mouse CD25 antibody, Clone PC61, BioLegend) every second day for 8 days. Similarly, to validate the inflammatory and apoptotic effects of caspase-14 in IBD mice, caspase-14 activity was neutralized by administering 100 µg of anti-caspase-14 (MBS668936, Clone: ABM1B24, MyBioSource) via an i.p. route. Experimental colitis was induced by addition of 4% (wt/vol) DSS in the drinking water for 7–9 days before sacrifice for all experimental groups.

### 2.10. Protein Extraction and Western Blotting

Colon tissue was lysed using a protein quantification kit (BCA Assay, Cat # KT3001, Abbkine) according to the manufacturer’s instructions (Bio-Rad, Hercules, CA, USA) with a homogenizer (Model 200 Double insulated; PRO Scientific Inc., Oxford, CT, USA). The total protein concentration was measured by a DC^TM^ protein assay (Bio-Rad, USA). Proteins (20–30 μg) were mixed with 4× sample buffer (60 mM Tris-HCl pH 6.8, 25% glycerol, 2% SDS, 14 mM β-mercaptoethanol, and 0.1% bromophenol blue); boiled; separated with 12.5%, 10%, and 7.5% (*w/v*) sodium dodecyl sulfate-polyacrylamide gel electrophoresis; and transferred onto polyvinylidene difluoride membranes (Millipore, Billerica, MA, USA) using Trans-Blot Turbo (Bio-Rad). Each membrane was blocked with 5% nonfat dry milk in PBS with 0.1% Tween 20 (PBST, Sigma-Aldrich) for 1 h at room temperature. After overnight incubation at 4  °C with primary antibodies ((TNF-α; sc-1350, Santa cruz), (IL-17A; NBP1-42772, Novus biologicals) (IFN-γ; sc-373727, Santa cruz) (Claudin-1; ab15098, Abcam), (IL-10; ab34843), (TGF-β1; BS91338, Bioworld)), membranes were washed with Tris-buffered saline containing Tween 20 (20 mM Tris-HCl, 150 mM NaCl, 0.1% Tween 20) and then incubated with the secondary antibody (immunoglobulin G antibody conjugated with horseradish peroxidase, Cat #170-6516 Bio-Rad) for 1 h at room temperature. β-actin (sc-47778, Santa Cruz) was used as an internal control. Immunoreactive bands were detected using West-Q Pico ECL Solution (GenDEPOT, USA) and analyzed by FluorChem E (Hybrid HY8300, San Jose, CA, USA). Band intensity was quantified using ImageJ 2.1.4.6 software (National Institutes of Health, Bethesda, MD, USA). 

### 2.11. cDNA Synthesis and Real-Time PCR

Total RNA (1 µg) was extracted from mice colon samples using Trizol reagent (Invitrogen, Carlsbad, CA, USA) according to the manufacturer’s instructions, and cDNA was purified using a minikit RNA and converted to cDNA using random hexamers (EcoDry^TM^ Premix, Takara, Clontech, USA). Real-time PCR (Applied Biosystems, Foster City, CA, USA) was performed using an iQ SYBR Green Supermix kit (Bio-Rad, Hercules, CA, USA). The oligonucleotide primers (Cosmo Genetech, Seoul, Korea) designed to analyze gene expression levels in mice using GAPDH as a house-keeping gene are shown in Table 1.

### 2.12. High-Throughput Sequencing and Transcriptomic Analysis

To construct the sequencing libraries, 500 ng of total RNA was used for complementary deoxyribonucleic acid (cDNA) synthesis. The RNA libraries were constructed using a SENSE 3′ messenger RNA (mRNA)-Seq Library Prep Kit (Lexogen Inc., Vienna, Austria), according to the manufacturer’s protocol. Gene expression was determined by high-throughput sequencing using a NextSeq 500 (Illumina Inc., USA), with single-end 75 sequencing using a NextSeq 500 by e-Biogen (Seoul, Korea). Differentially expressed genes were selected as those that showed a 2-fold increase or 0.5-fold decrease compared with the control. Gene clusters (hierarchical clustering), KEGG pathway, and heat maps were constructed based on MeV 4.9.0. The network and regulatory effects were analyzed using Ingenuity Pathway Analysis (IPA) by e-Biogen (Seoul, Korea). 

### 2.13. Statistical Analysis 

All statistical data from triplicate reactions were analyzed using GraphPad prism 8.0 software (GraphPad Software, San Diego, CA, USA). All values in the figures are expressed as mean  ±  standard error of the mean (SEM). Each experiment was performed in triplicate. Statistical comparison was performed using one-way analysis of variance (ANOVA, Tukey multiple comparison test). Each circle on the bar graph represents one mouse, and 4 to 7 mice/group were used in the experiment. Statistically significant differences were defined as * *p* ≤ 0.05; ** *p* ≤0.01; *** *p* ≤ 0.001 and **** *p* ≤ 0.0001.

### 2.14. Ethical Considerations

This study was carried out in accordance with the principles of the Guide for the Care and Use of Laboratory animals of Sungkyunkwan University and approved by the Institutional Animal Care and Use Committee at Sungkyunkwan University (SKKUIACUC2020-04-12-2), South Korea. All procedures described were performed in accordance with recommendations outlined in the National Institutes of Health Guide for the Care and Use of Laboratory Animals in accordance and guidelines of the Korean Animal Protection Law.

## 3. Results

### 3.1. ∆Pep27 Immunization Attenuated Proinflammatory Cytokines and Oxidative Stress in the DIC

System biology analyses in lung samples predicted that ∆pep27 could inhibit gastroenteritis and gut abnormalities (Figure 1A). To validate these findings, a DIC model was investigated (Figure 1B). DSS treatment aggravated clinical symptoms with a high disease activity index (DAI) score comprising of body weight loss, stools with signs of rectal bleeding, and colon shortening. However, Δpep27 alleviated DIC, restored body weight loss, decreased the DAI score, rescued the inflammation-related reduction of colon length, and limited colon damage (Figure 1C,E). As DSS disrupts the integrity of the mucosal barrier [21], a histopathological analysis using H&E staining was performed. The DIC group exhibited loss of epithelial crypts and disruption of tissue architecture compared to the normal control group (Figure 1F,H). Moreover, DIC led to an increase in cell death as depicted by the TUNEL assay. All these changes were significantly alleviated by ∆pep27, with reduced histological and apoptotic score (Figure 1G,I). DIC-induced proinflammatory and inflammatory transcripts comprising TNF-α and IL-17A were analyzed by qPCR. However, ∆pep27 significantly repressed these transcripts and protein levels (Figure 1J,K and Figure 2I,J) as previously described [20]. To check whether ∆pep27 itself might affect pathology and gene expression, we examined pathological changes and gene expression after ∆pep27 immunization without DSS treatment. However, ∆pep27 alone did not affect any of these genes involved in pro-inflammatory and anti-inflammatory genes nor pathological changes (HE staining and TUNEL assays), demonstrating that ∆pep27 immunization alone was similar to the normal control (Figure 1C–Q). Therefore, the ∆pep27 alone group was not included in further studies.

Upon tissue damage, ROS are generated as a defense mechanism against invading bacteria [28], and overproduction of ROS and RNS can exacerbate IBD symptoms [1]. To further corroborate inhibition of ROS and RNS production, myeloperoxidase (MPO) and malonaldehyde (MDA) levels were determined. ∆pep27 significantly inhibited DIC-induced MPO and MDA levels (Figure 1L,M), suggesting a reduction in neutrophil infiltration. ∆pep27 immunization substantially attenuated IBD symptoms in DSS-treated mice; nonetheless, ∆pep27 immunization alone did not show any pathological symptoms, i.e., the disease activity index parameters and colon length were the same as normal (Figure 1C,E). This notion was further confirmed by investigating the inflammatory cytokines TNF-α, IL-17A, oxidative stress parameters (MPO and MDA), and histological and TUNEL assay findings in the ∆pep27-only group were similar to those of the control (normal) mice (Figure 1C,Q). Collectively, Δpep27 alone did not induce any pathological symptoms. Thus, the effect of ∆pep27 was investigated by comparing DSS and ∆pep27-DSS with the control group. 

### 3.2. ∆Pep27 Immunization Diminished Proinflammatory Cytokines, and Caspase-14 Expression via Treg Upregulation to Prevent Barrier Disruption

An insight into the system biology analysis configures the protective role of ∆pep27 to mitigate the dysregulated immune response and aberrant cytokines production via antioxidant genes induction and regulating signaling pathways in colon (Figure 2A). Transcriptome analyses showed that the caspase-14 gene, a member of the caspase-1 family responsible for cytokine activation and apoptosis in keratinocytes [29,30], was significantly upregulated but not defined in IBD (Figure 2A). ∆pep27 dose-dependent activity suggested that three repeats of ∆pep27 immunization significantly attenuated DIC-induced proinflammatory and inflammatory mediators comprising TNF-α, IL-17A, and caspase-14 at mRNA and protein levels (Figure 1J,K and Figure 2B,I,K), indicating repression of inflammation by ∆pep27.

As dysfunction of Tregs or impairment of IL-10 and TGF-β1 signaling in Tregs led to IBD [5,6,7], cytokines correlated with Treg transcription factors were examined by qPCR. Compared with the DIC group, ∆pep27 increased anti-inflammatory transcripts and proteins of Treg relevant factors comprising FOXP3 (Treg), IL-10, and TGF-β1 (Figure 2C,E,M,O), indicating immune tolerance induction by ∆pep27.

### 3.3. DIC-perturbed Biodiversity of Gut Microbiota Is Restored by ∆Pep27

IBD leads to dysregulation of gut microbiota [28]. Thus, we sought the remedial effect of ∆pep27 on DIC, and the microbiome taxonomic profile was analyzed by 16S rRNA pyrosequencing. Compared with the control at the phylum level, DIC increased the relative abundance of Bacteroidetes whereas the relative abundance of Firmicutes and Verrucomicrobia was decreased (Figure 3A). Interestingly, ∆pep27 restored the perturbed family abundance including that of Lachnospiraceae, Lactobacillaceae, and Ruminococcaceae (Figure 3B). At the species level, DIC resulted in a high abundance of *Bacteroides vulgatus* but a low abundance of *Lactobacillus murinus* and *Akkermansia muciniphila*. However, ∆pep27 restored the relative abundance of dysbiotic flora, similar to the control (Figure 3C). In the DIC group, the relative abundance of alpha diversity indices such as operational taxonomic units (OTUs), Chao1, Shannon, and Simpson diversity used to compare species richness, abundance and structural differences among the samples were collectively reduced compared with the control group. Nonetheless, the ∆pep27-DIC group evidently enhanced alpha diversity compared with the DIC group (Figure 3D,G), indicating that ∆pep27 immunization leads to an increase richness and diversity of gut microbiota, which ameliorated inflammation associated taxa, and hence the low bacterial biodiversity of DIC.

### 3.4. Treg Depletion Is Restored by ∆Pep27 to Prevent Inflammation and Apoptosis

The effectiveness of anti-CD25 to deplete Tregs has been widely reported using different rodent models [26,27]. Tregs were depleted in the WT and DSS groups by 76.5% and 42.9% after neutralization with α-CD25 antibody compared with non-depleted WT and DSS groups, respectively [27]. α-CD25 treatment exacerbated the IBD symptoms more profoundly in the DIC subjects with high inflammatory and apoptotic index accompanied by an increase in DAI score (Figure 4A,B,E). In our study, α-CD25 treatment in the DSS group decreased the FOXP3 transcript by 70% compared with DSS only, whereas α-CD25 treatment in ∆pep27+DSS group showed a decrease of 26.2% compared with the non-treated ∆pep27+DSS group (Figure 4M). Consistently, the CD25 protein levels after depletion in control, DSS, and ∆pep27+DSS were decreased by 31.12%, 48.48%, and 42.23% respectively compared with non-depleted control, DSS, and ∆pep27+DSS groups (Figure 4P). Consistent with the previous findings [26,27], we observed significant Treg-depletion of FOXP3 transcript and CD25 protein. Moreover, CD25 antibody seems to be specific for Treg, and did not affect other effector T cells [31]. We observed that the DIC-control had increased transcripts and protein levels of TNF-α, IFN-γ, and caspase-14, though these were further upregulated following α-CD25 treatment (Figure 4J,L,Q,S), suggesting CD25-dependent attenuation of inflammatory responses. In contrast, α-CD25 treatment in the ∆pep27-DIC group showed significantly reduced FOXP3, IL-10, and TGF-β1 transcripts compared with the untreated ∆pep27-DIC group (Figure 4M,O). At the protein level, α-CD25 treatment in the ∆pep27-DIC group showed significantly decreased CD25 and IL-10 levels compared with the untreated group, whereas TGF-β1 was marginally affected (Figure 4T,V). These observations are likely associated with the dynamic role of ∆pep27-upregulated Treg-relevant factors and subsequent limitation of α-CD25 neutralization. These results indicate that ∆pep27 attenuated DIC symptoms, inflammation, and cell death by via upregulation of Tregs.

### 3.5. Caspase-14 Depletion Uncovers Proinflammatory and Apoptotic Nature of Caspase-14

TNF-α and IFN-γ are major proinflammatory cytokines that synergistically drive epithelial barrier dysfunction and apoptosis, particularly during colitis [32]. Thus, to identify the role of caspase-14, a neutralization experiment was conducted (Figure 5A). Caspase-14 neutralization in the DIC group attenuated body weight loss and the DAI score (Figure 5A,B), and showed longer colon length than the non-neutralized group (Figure 5D,E), relieved inflammation, and cell death, as confirmed by H&E staining and TUNEL assay (Figure 5F,I), suggesting caspase-14 as an inflammatory and apoptotic marker. Consistently, ∆pep27 significantly attenuated DIC-induced TNF-α, IFN-γ and caspase-14 transcripts; however, ∆pep27 upregulated Treg-related gene transcripts such as FOXP3, IL-10, and TGF-β1 (Figure 5J,L). Interestingly, caspase-14 neutralization in the ∆pep27-DIC group revealed that neither anti-inflammatory (CD25, IL-10, and TGF-β1) nor proinflammatory genes (TNF-α and IFN-γ) were significantly affected (Figure 5), indicating that the ∆pep27-driven anti-inflammatory milieu dominates, regardless of caspase-14 neutralization. 

## 4. Discussion

Previously, ∆pep27 immunization provided substantial protection against pneumococcus at the nasopharynx, regardless of the serotype [16,17]. Additionally, Δpep27 significantly attenuates inflammatory cytokines in lung infection [17,33]. Consistently, Δpep27 immunization increased the survival rate and macrophage viability during the resumption of disrupted alveoli by upregulating the SPRR family after pneumococcal challenge [20]. Δpep27 significantly increased the levels of the SPRR genes that are involved not only in the establishment of the physical barrier but also in cell migration and wound healing [34,35]. This results in a strengthened alveolar barrier and enhanced resistance to external stressors [35], thereby ameliorating gut inflammation and suggesting a robust regenerative and oxidant stress-relieving mechanism to re-establish immunological tolerance. Based on these novel findings, we evaluated microbial composition and neutralized the effect of critical markers to analyze the protective effect of ∆pep27 against experimental colitis.

System biology analysis predicted the inhibition of gut abnormality by ∆pep27. IBD instigates colitis symptoms characterized by inflammatory cell infiltration, secretion of proinflammatory cytokines, oxidative stress release, and subsequent barrier disruption in the colonic tissue [1,20]. However, these gut abnormality parameters were significantly alleviated by ∆pep27, suggesting Δpep27 potentially ameliorates histopathological feature and intestinal barrier disruption by augmenting tight junction expression. Thus, ∆pep27 seems to protect against IBD by inhibiting Th17 cells via Treg upregulation and an antioxidative stress-relieving mechanism and can be a highly pragmatic way to relieve gut inflammation. Additionally, ∆pep27 significantly inhibited oxidative stress parameters in DIC, consistent with the findings of ref. [36] where Korean red ginseng enhanced Δpep27 potency to suppress both ROS generation and ERK signalling-mediated cell death. 

Previously, it was reported that inflammatory cytokine-induced apoptosis might lead to a breakdown of epithelial barrier function, allowing invasion of harmful bacteria as well as inflammation [28]. Induction of Tregs using probiotics-defined bacteria is the key mechanism to confer health protection to the host. Gut microflora and their metabolites play either a protective or a detrimental role, and short-chain fatty acids such as acetate and butyrate augment the function of Treg cells [11,37]. Ruminococcaceae and Lachnospiraceae, two predominant families of Firmicutes in the human colon correlated positively with the expression of the gene-encoding Treg transcription factor FOXP3 or the gene-encoding the anti-inflammatory cytokine IL-10 or both [38]. Consistent with our results, DIC microbiota showed an increase of Bacteroidetes and a reduction of Firmicutes [39]. Bacteroides exhibited negative correlation with TJ proteins, while positive correlation with proinflammatory cytokines [40]. However, Δpep27 increased the abundance of Clostridia and Verrucomicrobia, which activates intestinal epithelial cells to induce FOXP3 and oppose colitis induction [11,37]. Comprehensively, DIC increased the relative abundance of pathogenic bacteria and led to colonic inflammation [39,40], while Δpep27 restored the dysbiosis pattern to normal at all phylogenetic levels.

Caspase-14 has been studied in the context of terminal keratinocyte differentiation and is thought to mediate DNA fragmentation during Treg apoptosis [41]. Previously, caspase-14 was classified as a member of caspase-1 family mainly expressed in cornifying epithelia and terminal keratinocyte differentiation. However, neither its transcriptional regulation nor substrate has been clarified [42]. We found significant upregulation of the caspase-14 as one of the contributing factors in the pathogenesis of IBD. In support of this, several inflammatory and oxidative stress models suggest collateral induction of caspase-3 and caspase-14 [43,44], thereby increasing cytokines level [30]. In the DIC model, caspase-3 is a well-defined apoptotic marker [45]. Consistently, Δpep27 inhibited Wnt5a expression via NFAT suppression, resulting in the downregulation of the expression levels of the proinflammatory cytokine TNF-α and caspase-14. We showed that Δpep27-DIC significantly ameliorated the levels of proinflammatory transcripts (TNF-α, caspase14, Wnt5a, and Wnt11) in both the VEET and VIVIT groups, indicating the significant role of NFAT suppression in ameliorating colitis. Δpep27 upregulated Tregs and increased the levels of anti-inflammatory transcripts (IL-10 in particular) in the VEET and VIVIT groups, suggesting that Δpep27 upregulated the Treg-dependent blockade of NFAT that can be an alternate therapeutic target in IBD [20]. Moreover, caspase-14 is considered as an inflammatory and apoptotic marker for cytokine activation in the chronic inflammatory Behcet’s disease [30]. Once caspase-14 has been biochemically characterized, it could be considered as an alternate IBD therapeutic target. 

We demonstrated the effect of ∆pep27 immunization on predominant genes in the gut dysbiosis that contributed to colitis progression. Although an increased rate of apoptosis was observed in the Treg-depleted model, ∆pep27 immunization increased Treg, rendering them functional. Collectively, our findings suggest that IBD in response to DSS induces proinflammatory cytokines and ROS, resulting in upregulation of caspase-14 expression. This instigates colitis symptoms characterized by infiltration of inflammatory cells, leading to Treg depletion and microbial dysbiosis and resulting in the loss of barrier integrity. In contrast, ∆pep27 immunization inhibited DSS-induced caspase-14 expression to attenuate experimental colitis via restoration of functional Tregs and healthy gut microbiota composition, suggesting a robust regenerative and antioxidant mechanism to re-establish immunological tolerance (Figure 6). Thus, ∆pep27 might act as a promising candidate therapy in clinical applications irrespective of dietary practices/determinants. Despite a novel anti-inflammatory mechanism suggested in experimental IBD mice, clinical trials of ∆pep27 in IBD patients warrant further research and exploration.

## Figures and Tables

**Figure 1 microorganisms-10-01871-f001:**
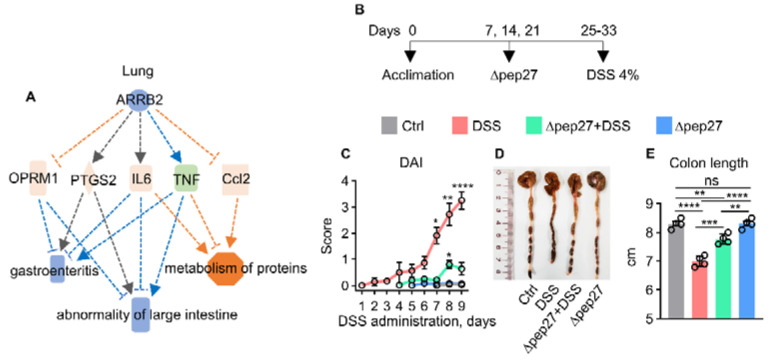
Δpep27 immunization attenuates DIC-induced IBD symptoms by suppressing inflammatory mediators and cell death via antioxidative mechanism. (**A**) System biology analysis of lung transcriptome data predicts that Δpep27 mitigates both gastroenteritis and abnormalities of the large intestine. (**B**) Experimental plan to induce colitis in mice. (**C**–**E**) Progression of colitis was evaluated by DAI (**C**), with representative colon images (**D**) and colon length measurement (**E**). (**F**,**H**) Colon sections were stained with H&E (20× magnification, scale bar, 200 µm) (**F**) with histological grading (**H**). (**G**,**I**) TUNEL assay (**G**) with statistical analysis (**I**) (200× magnification, scale bar, 100 μm). (**J**–**K**) mRNA expression level was analyzed by qPCR using GAPDH as an internal control. (**L**–**M**) Oxidative stress response as an assessment of reactive oxygen species (ROS) and reactive nitrogen species (RNS) was quantified by myeloperoxidase (MPO) (**L**) and malondialdehyde (MDA) (**M**) assays. (**N**–**Q**) Dose dependent activity of ∆pep27 immunization was determined by qPCR using GAPDH as an internal control. Statistically significant differences were defined as * *p* ≤ 0.05; ** *p* ≤ 0.01; *** *p* ≤ 0.001; and **** *p* ≤ 0.0001. Statistical comparison among groups was performed by mean ± SEM using one-way ANOVA followed by Tukey multiple comparison test (n = 4–5 mice/group). Results are representative of three independent experiments.

**Figure 2 microorganisms-10-01871-f002:**
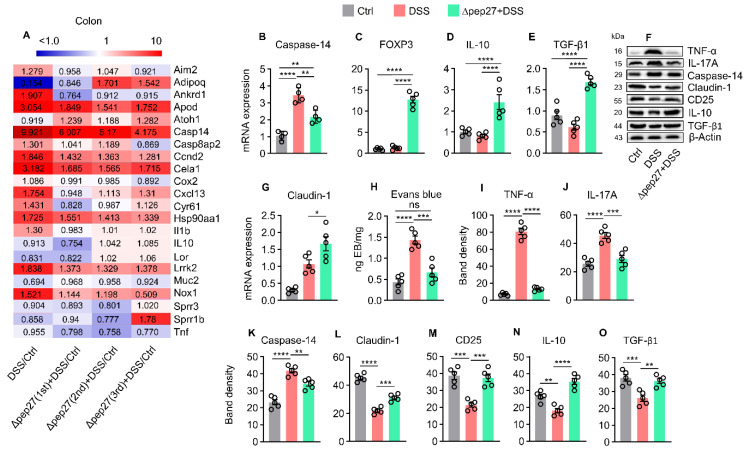
∆pep27 immunization repressed proinflammatory cytokines expression via Tregs upregulation to enhance barrier permeability. (**A**) Heat map analysis revealed inhibition of inflammation-related genes, antioxidant genes, and repression of the caspase-14 by ∆pep27 in colon. (**B**–**F**) Caspase-14 transcription (**B**), Treg transcription factor and anti-inflammatory mediators (**C**–**E**) in DIC were determined by qPCR. (**F**) Levels of various proteins were confirmed by western blot analysis using β-actin as an internal control. Band intensity was quantified using ImageJ 2.1.4.6 software (National Institutes of Health, Bethesda, MD, USA). (**G**,**H**) Epithelial barrier disruption was determined by tight junction gene (**G**) and Evans blue dye assay (**H**). (**I**–**O**) Protein expression was quantified by band density and analyzed by ImageJ 2.1.4.6 software (National Institutes of Health, Bethesda, MD, USA). Statistically significant differences were defined as * *p* ≤ 0.05; ** *p* ≤ 0.01; *** *p* ≤ 0.001; and **** *p* ≤ 0.0001. Statistical comparison among groups was performed by mean ± SEM using one-way ANOVA followed by Tukey multiple comparison test (n = 5 mice/group). Results are representative of three independent experiments.

**Figure 3 microorganisms-10-01871-f003:**
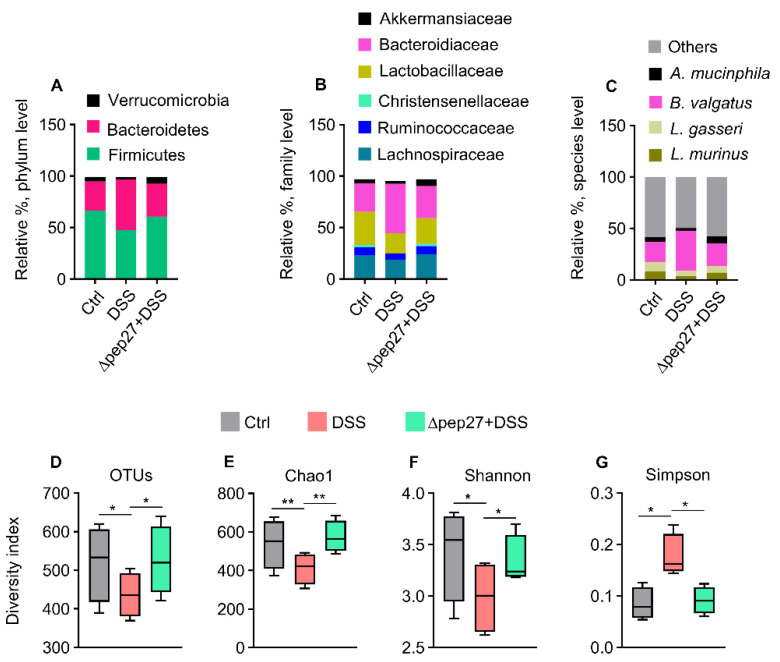
∆pep27 immunization restores healthy gut microbiota and species diversity. (**A**–**G**) Effect of ∆pep27 on the microbial composition at various taxonomic levels in DSS-induced colitis (DIC) mice. Fecal pellets were analyzed by pyrosequencing of the 16S rRNA. Relative abundance (%) at phylum (**A**), family (**B**), and species levels (**C**). Microbiome taxonomic profile revealed that the major phyla of *A. mucinphila*, *B. valgatus*, *L. gasseri* and *L. murinus* constitute approximately 40% of the whole bacterial population at species level (**C**). Diversity indices of gut microbiota were evaluated as (**D**) operational taxonomic unit (OTU) counts, (**E**) Chao1, (**F**) Shannon, and (**G**) Simpson in the correspondent groups. Statistically significant differences were defined as * *p* ≤ 0.05; ** *p* ≤ 0.01 as significant. Statistical comparison among groups (n = 4 mice/group) was performed by mean ± SEM using one-way ANOVA followed by Tukey multiple comparison test.

**Figure 4 microorganisms-10-01871-f004:**
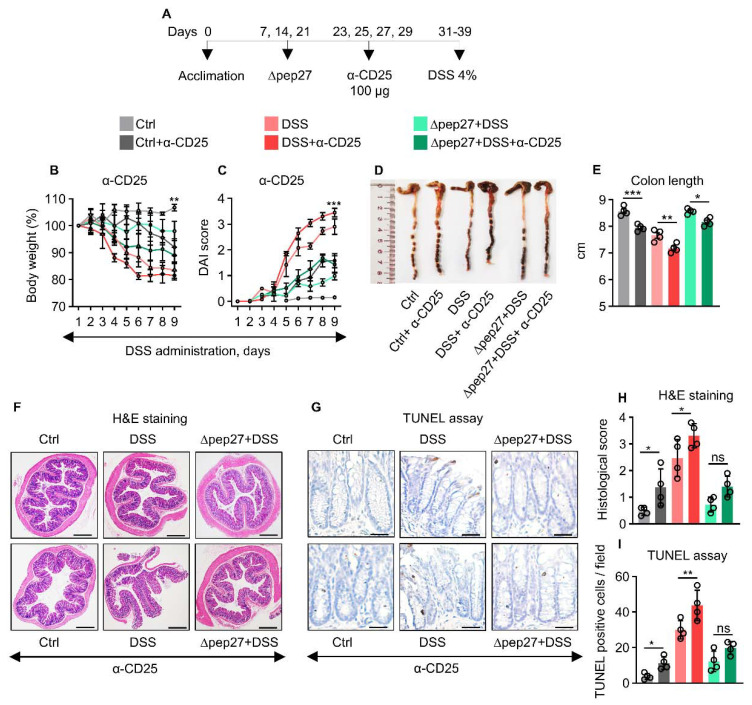
∆pep27 immunization restores Treg depletion to prevent DIC-induced inflammation and apoptosis. (**A**–**E**) Neutralization protocol of Treg with anti-CD25 (**A**). Body weight (**B**), DAI score (**C**), and colon length with their representative images (**D**,**E**) after Treg neutralization using CD25 antibody. (**F**,**G**) Colon samples were H&E stained (**F**) and analyzed by TUNEL assay (**G**). (**H**) Representative photomicrographs of H&E staining at 20× magnification, scale bar = 200 µm, and (**I**) TUNEL assay after anti-CD25-treatment (200× magnification, scale bar = 100 μm). (**J**–**O**) Transcripts in the colon samples were analyzed by qPCR. (**P**) Protein levels were confirmed by western blot analysis. (**Q**–**V**) Band densities were analyzed by ImageJ 2.1.4.6 software (National Institutes of Health, Bethesda, MD, USA). Data represent two independent experiments and are presented as the mean ± SEM. Statistically significant differences were defined as * *p* ≤ 0.05; ** *p* ≤ 0.01; *** *p* ≤ 0.001; and **** *p* ≤ 0.0001 (n = 4 mice/group).

**Figure 5 microorganisms-10-01871-f005:**
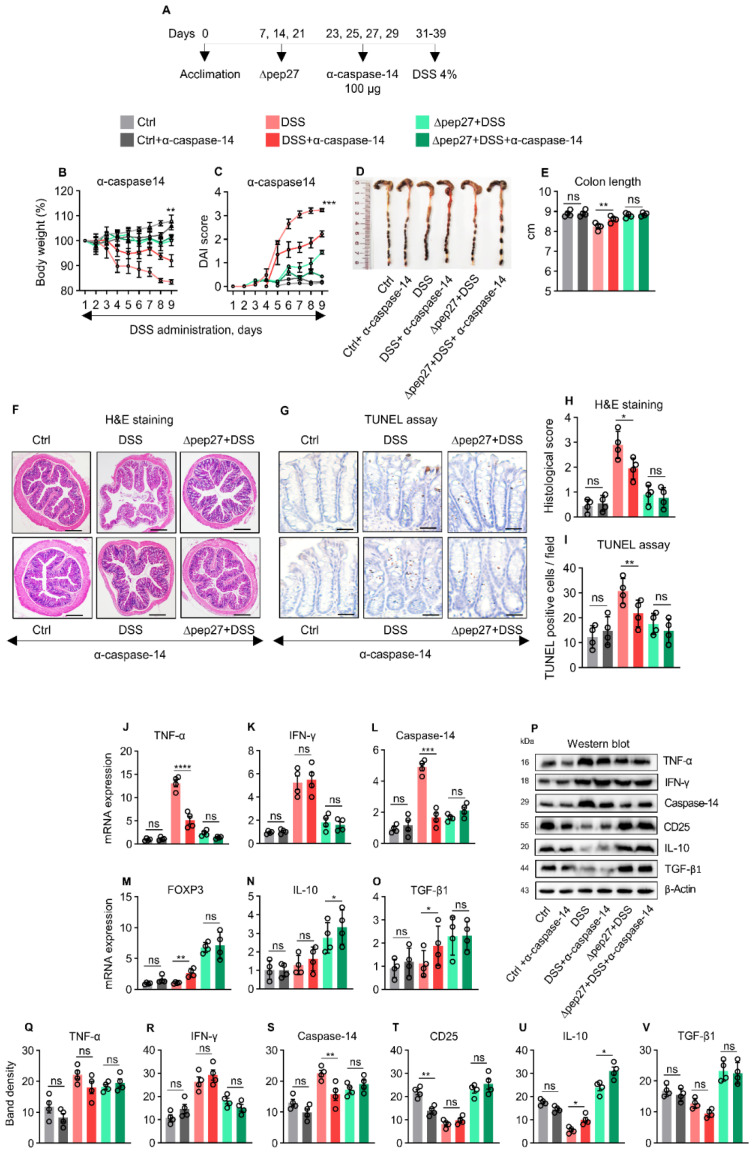
∆pep27 represses caspase-14 via Treg upregulation and alleviates DIC. (**A**–**E**) Blockade of caspase-14 expression via anti-caspase-14 neutralization assay (**A**). Body weight (**B**), DAI score (**C**), with representative colon images (**D**) and colon length (**E**) after anti-caspase-14 treatment. (**F**,**G**) Colon samples were H&E stained (**F**) and analyzed by TUNEL assay (**G**). Representative photomicrographs of H&E staining at 20× magnification, scale bar = 200 µm and histological score (**F**,**H**) and TUNEL assay (200× magnification, scale bar = 100 μm) after anti-caspase-14-treatment (**G**,**I**). (**J**–**O**) Transcripts in the colon samples were analyzed by qPCR. (**P**) Protein levels were confirmed by western blot analysis. (**Q**–**V**) Band densities were analyzed by ImageJ 2.1.4.6 software (National Institutes of Health, Bethesda, MD, USA). Data represent two independent experiments and are presented as the mean ± SEM. A * *p* value of < 0.05 is considered as statistically significant (n = 4 mice/group). Statistically significant differences were defined as * *p* ≤ 0.05; ** *p* ≤ 0.01; *** *p* ≤ 0.001; and **** *p* ≤ 0.0001.

**Figure 6 microorganisms-10-01871-f006:**
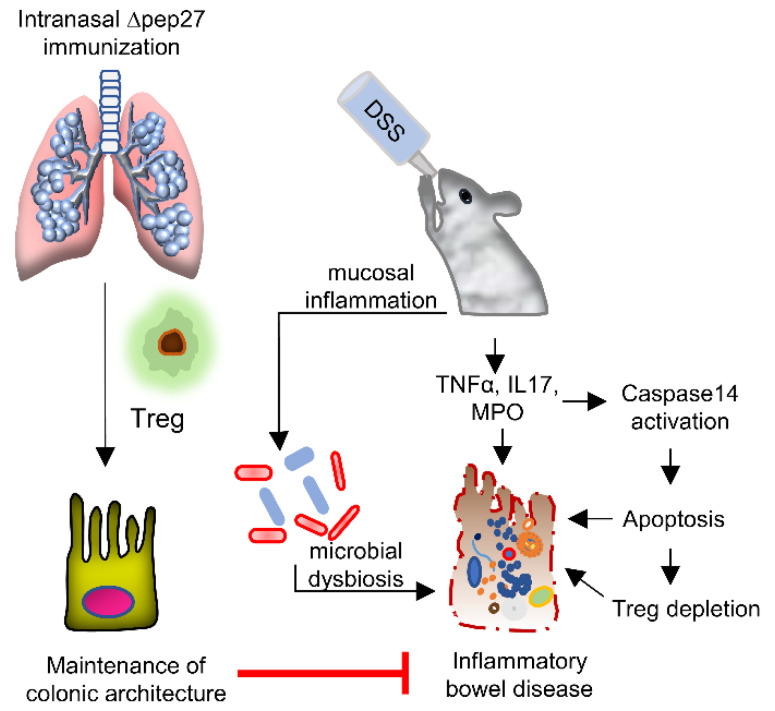
∆pep27-elicited-Treg restores mucosal immune response in DIC. DIC depletes Tregs and subsequently induces proinflammatory cytokines, caspase-14, resulting in inflammation and apoptosis as well as gut microbiota disruption. In contrast, ∆pep27-dependent Treg upregulation elicits robust anti-inflammatory mucosal tolerance and maintains the barrier integrity for intestinal immune homeostasis and subsequent restoration of gut microbiota.

**Table 1 microorganisms-10-01871-t001:** Primers used in this study.

Gene Name		Sequence (5′ to 3′)
Caspase-14	F	GATGAGGTTGCTGTGCTCAA
	R	TTGGTTTTCCTTCCTGCATC
Claudin-1	F	GATGTGGATGGCTGTCATTG
	R	CCACTAATGTCGCCAGACCT
FOXP3	F	CCTGCCTTGGTACATTCGTG
	R	TGTTGTGGGTGAGTGCTTTG
IFN-γ	F	ACAGCTCTCCGTCCTCGTAT
	R	CCTTAGAGTAGAAAGACCG
IL-10	F	AGCCACATGCTCCTAGAGC
	R	GCCTGGGGCATCACTTCTAC
IL-17A	F	TCCAGAAGGCCCTCAGACTA
	R	AGCATCTTCTCGACCCTGAA
TGF-β1	F	ACCATGCCAACTTCTGTCTG
	R	CGGGTTGTGTTGGTTGTAGA
TNF-α	F	GCC TCTTCTCATTCCTGCTT
	R	TGGGAACTTCTCATCCCT TTG
GAPDH	F	TCAACAGCAACTCCCACTCTTCCA
	R	ACCCTGTTGCTGTAGCCGTATTCA

## Data Availability

The gene expression analysis data for colon tissue is deposited in the National Center for Biotechnology Information database (GEO accession number GSE139916) (http://www.ncbi.nlm.nih.gov/geo/, accessed on 2 January 2020) by e-Biogen (Seoul, Korea). The sequence data for microbiome analysis are available in the NCBI (https://dataview.ncbi.nlm.nih.gov/object/PRJNA714775, accessed on 2 January 2020) SRA database (SRA: SRR13973109).

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
