# Peer review of "Pep27 Mutant Immunization Inhibits Caspase-14 Expression to Alleviate Inflammatory Bowel Disease via Treg Upregulation"

_microorganisms, 2022, doi:10.3390/microorganisms10091871_

Round 1
Reviewer 1 Report
Congratulations on a well-designed and conducted study. In my opinion, it deals with a topic that has been little known so far and is an interesting contribution to human clinical trials.
I have no significant comments on the content of the article.
My suggestion is only to replace the word microflora with microbiota in verse 38.
Author Response
Authors appreciate the referees for understanding and acknowledging the impact of our research by providing us with valuable suggestions to undoubtedly enhance the credibility of our findings. We hope that our explanation to address the referee’s concerns will satisfy the reviewer in the revised manuscript.
Ans. As per reviewer instruction, the term microflora has been replaced to microbiota.
Reviewer 2 Report
The authors herein applied a compact experimental design to test the therapeutic potential of intranasal immunization of pneumococcal pep27 mutant (Δpep27) against the development of acute chemical injury to the large bowel. The authors indeed, show that ∆pep27 immunization ameliorated DSS-induced colitis and protected the mice from severe injury. In an effort to reveal the underlying mechanisms for this protective effect they did transriptomic and macrobiotic analyses and report that ∆pep27 immunization ameliorated pro-inflammatory gene expression whereas, concomitantly restored functional Tregs. In parallel, a healthy gut microbiota composition was restored in immunisation-treated mice, the end result being the redevelopment of an intact epithelial barrier, which was deranged by the administration of DSS. the authors propose that a pivotal mechanism was repression of cascade-14 activity by the restored Tregs.
The work is well executed and the results are well-presented.
The major critique I have with this submission is that the murine model that was used is very little relevant to IBD. DSS colitis is a model of acute injury and repair and is distinct from the chronic inflammation that takes place in IBD. the authors should change the title of the article because it is misleading for this reason. they should also highly soften their statements about the relevance and therapeutic potential of Δpep27 immunization in IBD. it is worth mentioning that the comics analysis from their previous work pointed to an association with gastroenteritis which is an acute self-limited phenomenon.
Secondly, the authors should explore more, at leat in their discussion about the relevance of an intranasal immunisation and protection from inflammation in another organ. Where the protective Tregs were produced and how did they arrive at the inflamed bowel? are there specific trafficking molecules that interconnect nasal and gut mucosae?
Author Response
Authors appreciate the referees for understanding and acknowledging the impact of our research by providing us with valuable suggestions to undoubtedly enhance the credibility of our findings. As per reviewer statement, we have addressed all your comments in order to enhance the quality of the manuscript. revised the terms and expressions. We hope that our explanation to address the referee’s concerns will satisfy the reviewer in the revised manuscript.
